# Observation of formation and local structures of metal-organic layers via complementary electron microscopy techniques

Xinxing Peng[1,5], Philipp M. Pelz [1,2,5], Qiubo Zhang [3,5], Peican Chen[4], Lingyun Cao[4], Yaqian Zhang[1,2], Hong-Gang Liao [4] ✉, Haimei Zheng [2,3], Cheng Wang [4], Shi-Gang Sun[4] & Mary C. Scott [1,2] ✉

Metal-organic layers (MOLs) are highly attractive for application in catalysis, separation, sensing and biomedicine, owing to their tunable framework structure. However, it is challenging to obtain comprehensive information about the formation and local structures of MOLs using standard electron microscopy methods due to serious damage under electron beam irradiation. Here, we investigate the growth processes and local structures of MOLs utilizing a combination of liquid-phase transmission electron microscopy, cryogenic electron microscopy and electron ptychography. Our results show a multistep formation process, where precursor clusters first form in solution, then they are complexed with ligands to form non-crystalline solids, followed by the arrangement of the cluster-ligand complex into crystalline sheets, with additional possible growth by the addition of clusters to surface edges. Moreover, high-resolution imaging allows us to identify missing clusters, dislocations, loop and flat surface terminations and ligand connectors in the MOLs. Our observations provide insights into controllable MOL crystal morphology, defect engineering, and surface modification, thus assisting novel MOL design and synthesis.

Metal-organic frameworks (MOFs) are organic-inorganic hybrid crystalline porous materials that consist of diverse metal nodes and organic linkers[1,2]. In the last two decades, MOFs have attracted widespread attention for applications in catalysis, sensing, chemical separation, gas capture and energy storage because of their tunable porous structure and high specific surface areas[3–5]. Metal-organic layers (MOLs), also known as 2D MOF sheets, have attracted considerable attention because of their inherited features of both 2D materials and MOFs[6].

Despite the broad utility and interest of MOLs/MOFs, there have been few reports of in-situ observations of their nucleation and growth in solution, largely due to the absence of effective characterization techniques. To date, the most prevalent proposed formation process of MOLs/MOFs is the classic nucleation and growth model, in which the monomer-by-monomer addition mechanism is reported to play a role in the crystal formation process[7,8]. However, whether classical mechanisms can be applied to the formation of MOLs/MOFs is still

[1]National Center for Electron Microscopy, Molecular Foundry, Lawrence Berkeley National Laboratory, Berkeley, California 94720, USA. [2]Department of Materials Science and Engineering, University of California, Berkeley, California 94720, USA. [3]Materials Science Division, Lawrence Berkeley National Laboratory, Berkeley, California 94720, USA. [4]State Key Lab of Physical Chemistry of Solid Surfaces, Collaborative Innovation Center of Chemistry for Energy Materials, College of Chemistry and Chemical Engineering, Xiamen University, Xiamen 361005, P. R. China. [5]These authors contributed equally: Xinxing Peng, Philipp M. Pelz, Qiubo Zhang. ✉e-mail: hgliao@xmu.edu.cn; mary.scott@berkeley.edu

under discussion[9,10]. Although different techniques, including in-situ spectroscopic techniques, liquid atomic force microscopy and theoretical simulations have been employed to study the formation of MOFs[11–13], it is still challenging to obtain a complete picture of the MOL formation process, especially the early-stage growth. Due to the lack of comprehensive understanding of MOL growth mechanisms in solution, it is difficult to precisely control their growth and structure[14–16]. Liquid-phase transmission electron microscope (LPTEM) has been a revolutionary technique in the characterization of materials, enabling the direct visualization of dynamic processes in a liquid phase with high spatial resolution[17]. Therefore, LPTEM holds the potential to provide rich information on the formation of MOLs.

In addition to formation mechanisms, there is a need for a better understanding of the structure-property relationships in MOLs/MOFs. Single crystal and powder X-ray diffraction techniques are usually used to determine the average and periodic structures of MOLs/MOFs[18]. However, local, non-periodic structural information, such as defects and interface structures, cannot be obtained by using these techniques. High-resolution TEM is the most powerful technique for imaging non-periodic local structures of materials[19,20]. Unfortunately, MOLs/MOFs are sensitive to electron beam irradiation and their crystalline structure cannot survive during typical TEM imaging[21–24], with a recent report identifying two distinct stages of structural changes in MOFs under electron beam irradiation[25]. Reducing the electron beam effect while maintaining high spatial resolution is an active area of research. Recent advances in imaging techniques of electron microscopy have been reported to show great potential for the characterization of beam-sensitive materials[26–28]. Techniques including cryogenic electron microscopy (cryo-EM), electron ptychography, the integrated differential phase-contrast technique and low dose imaging techniques based on the direct-detection electron-counting camera have been successfully used to characterize the bulk, surface/interface structure, and defects of MOFs with sub-unit-cell resolution[20,28–34]. These discoveries have greatly enriched our understanding of MOLs/MOFs.

To date, almost all reported results use 3D MOFs as a model system to study the local structures[35–37]. Nevertheless, high-resolution characterization of 3D MOFs is very challenging for two reasons: (1) The manual method to align a zone axis through iterative toggles between the diffraction and imaging modes often causes severe beam damage to the sample[28]. (2) The periodic and aperiodic structures we observe are projections on a 2D plane. Therefore, overlapping local structures can be ambiguous[38]. However, the 2D structures utilized in this study are naturally well-aligned for high-resolution imaging, and various defects and surface/interface structures can be easily identified[19,39].

In this work, we investigate the growth processes and image detailed local structures for beam-sensitive Hf-MOLs utilizing complementary TEM techniques, including LPTEM, cryo-EM, high-resolution TEM (HRTEM), high-angle annular dark-field scanning transmission electron microscopy (HAADF-STEM) and electron ptychography. Our results show that the MOLs undergo a multistep formation process, where precursor clusters are first generated in solution, then are complexed with ligands to form non-crystalline solids, followed by the arrangement of the cluster-ligand complex into crystalline sheets. Moreover, the high-resolution imaging reveals local structure in the MOL sheet, including missing clusters and disordered regions. We also image loop and flat surface terminations at MOL sheet edges. High-resolution ptychography allows us to identify the surface ligand layer and observe the structure of the ligand linker within the MOL.

## Results
### Synthesis and structural characterization of Hf-MOLs
In this work, we utilize a previously reported strategy to synthesize MOLs based on the coordination preferences between the inorganic nodes with organic linkers via bottom-up assembly[40]. The MOLs are composed of Hf (IV)-based inorganic nodes and organic ligands. The structure of Hf-based MOLs (Hf-MOLs) is built from the linkage of $[Hf_6(\mu_3\text{-}O)_4(\mu_3\text{-}OH)_4(carboxylate)_{12}]$ clusters and benzene-1,3,5-tribenzoate (BTB) ligands. The $Hf_6$ cluster is normally twelve coordinated to form 3D frameworks with high chemical and thermal stability, such as UiO-66 (Hf)[41]. To obtain 2D Hf-MOLs, we blocked six connection sites using formate groups such that the remaining six connection sites are in the same plane[40]. The growth solution was prepared by dissolving $HfCl_4$ and $H_3BTB$ in a mixture solvent of HCOOH, DMF and $H_2O$ (see experimental sections for details). We tracked the ex-situ growth process of Hf-MOLs by heating the growth solution for different amounts of time. Consistent with previous observations, the solution became opaquer and cloudier as the heating time increased. In the first 10 mins, the growth solution appeared colorless and transparent. At 12 mins, the solution started to turn white, and it became a white suspension after 17 mins (Supplementary Fig. 1a). The TEM results show that even if the solution was heated for only 30 mins, crystalline Hf-MOLs can be obtained (Supplementary Fig. 1b, c). We note that leaving the growth solution at room temperature for 2 months also results in a similar reaction product. Therefore, we infer that heating only changes the kinetics of Hf-MOL formation and it will not change the type of the product.

Figure 1a illustrates the formation of Hf-MOLs: the six-connected $Hf_6$ clusters coordinate with the three-connected BTB ligands to form a 2D network. The 3D MOFs can be synthesized by adjusting the ratio of $H_2O$ and HCOOH[40,42]. The structure of 3D MOFs[43,44] is shown in Supplementary Fig. 2. We first characterized the morphology, bulk structure, and composition of Hf-MOLs using HAADF-STEM. Considering MOLs materials are easily damaged by the electron beam, the specimens were imaged at cryogenic temperature to reduce the beam effect (Supplementary Fig. 3). STEM-EDS mapping was achieved using the Bruker Super-X EDS with four windowless silicon drift detectors, which have a solid angle of 0.7 steradian enabling high count rates with minimal dead time. The HAADF-STEM characterization and Super-X EDS analysis of a thin area are shown in Fig. 1b, c. The STEM image reveals that the MOLs are wrinkled nanosheets with areas as large as $1 \times 1\,\mu m^2$. The synthesized MOLs exhibit surface corrugation and crumples, which are also frequently found in other 2D materials, such as graphene[45]. The STEM-EDS mapping and spectra confirm the existence and uniform distribution of Hafnium, carbon, and oxygen within Hf-MOLs. The HAADF-STEM images are formed by high angle, incoherently scattered electrons, where the contrast is dominated by scattering from the heavy Hf atoms[46,47]. Therefore, the bright spots observed in Fig. 1c correspond to Hf-containing clusters. To confirm the layered structure of Hf-MOLs, tapping-mode AFM was conducted to measure the thickness of the Hf-MOLs. Figure 1d shows AFM images of a large area of a typical Hf-MOLs nanosheet. The AFM results in Fig. 1f indicate the thickness of synthesized Hf-MOLs films in Fig. 1d, e are 1.4 nm and 2.9 nm, which correspond to monolayer and bilayer, respectively. Besides monolayers and bilayers, some regions with a thickness of 4.2 nm were also observed, which correspond to a multilayer structure. (Supplementary Fig. 4).

### Investigation of MOL initial growth using liquid-phase TEM and Cryo-EM
The in-situ observation of Hf-MOL growth in liquids was performed using a custom-made ultrathin silicon nitride chip with $50 \times 200\,\mu m^2$ viewing area and 10 nm thick membrane. Typically, Hf-MOL formation takes place under solvothermal conditions at a high temperature (120 °C). Since the custom-made liquid chips cannot be directly used for heating, the growth solution was first heated at 120 °C for 10 min in glass vials. Then, the growth solution was injected into the liquid chips and sealed for TEM observation. We first observed the formation of Hf-clusters in the solution (Supplementary Movie 1), visible as uniform dark spots shown in the TEM images. Few clusters are observed at the beginning. As the reaction progresses, the number of Hf-cluster

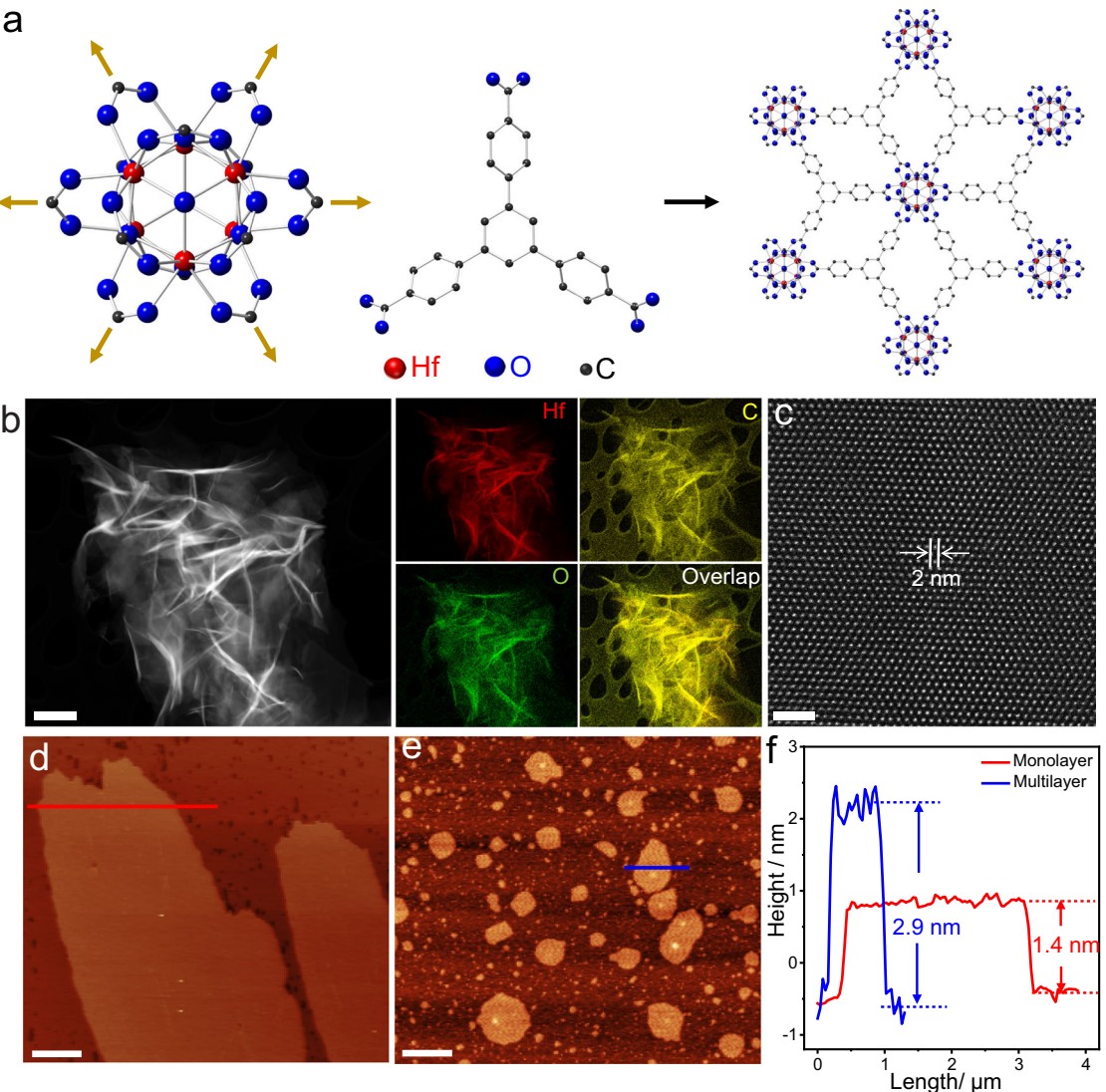

**Fig. 1 | The bulk structure characterization and thickness measurement of Hf-MOLs nanosheets. a** Schematic illustration of the construction of MOLs from 6-connected Hf-clusters and 3-connected BTB ligands with the connectivity indicated by golden arrows. The elements are represented as Hf, red; O, blue; C, black. **b** Representative HAADF-STEM image and the corresponding elemental maps of Hf, C and O using Super-X EDS at cryogenic temperature. **c** A representative high-resolution HAADF-STEM image of MOLs shows hexagonal arrangements of clusters. **d**, **e** Tapping-mode AFM topography of the Hf-MOLs with different regions. **f** Height profile along the red line in d and blue line in **e**. Scale bars: **b** 500 nm; **c** 10 nm; **d**, **e** 1 μm.

increases, with >50 clusters detected at 218.2 s in the same viewing area (Supplementary Fig. 5). Therefore, the first stage is the growth of the Hf-clusters, and the cluster formation occurs before the formation of Hf-MOLs. We further confirm this by HRTEM characterization. HRTEM images show that there are many independent clusters in the sample synthesized ex-situ (Supplementary Fig. 6). Although small cluster formation was captured, the motion of clusters is very slow due to strong attachment to the silicon nitride window (Supplementary Fig. 7 and Supplementary Movie 2). Clusters not attached to the window were not visible due to their small size and low contrast in the liquid bulk. To achieve the self-assembly of clusters, we focused the electron beam to generate bubbles in the vicinity of regions with clusters, creating a flowing liquid and increasing the cluster mobility and density locally. Figure 2a shows the self-assembly process of Hf-clusters (Supplementary Movie 3). The transformation from disordered, low-density arrangements of clusters to a 2D close packing with areas of hexagonal arrangements was observed.

To further understand the formation mechanism of the Hf-MOLs, we characterized the early-stage product of Hf-MOLs (see

experimental sections for details) using cryo-EM. Figure 2b–d shows the HRTEM image of the Hf-MOLs synthesized at 120 °C for 30 min. HRTEM (left panel) and the corresponding Fast Fourier Transform (FFT) image (middle panel) of the selected area reveal that the center of the Hf-MOLs film presents a highly crystalline structure, as shown in Fig. 2c. The periodic structure is clearly observed in the inverse FFT (IFFT) image (right panel) in Fig. 2c. However, the edge region of the Hf-MOLs exhibits disorder from the enlarged image of the surface region, as shown in Fig. 2d. The crystalline core and disordered surface structure indicate that the nucleation process first happened in the center part of the thin film. An FFT of the disordered region (middle panel in Fig. 2d) indicates a characteristic distance between the clusters of 2 nm, which is the same as the distance between neighboring clusters of highly crystalline Hf-MOLs (Supplementary Fig. 8). Therefore, we prove that clusters are arranged randomly in the early stage, but adjacent clusters are connected by bridge ligands before the formation of crystalline Hf-MOLs. An IFFT image (right panel in Fig. 2d) of the surface region confirms clusters gather and keep a certain distance of 2 nm, showing that clusters dimers, trimers and multimers are

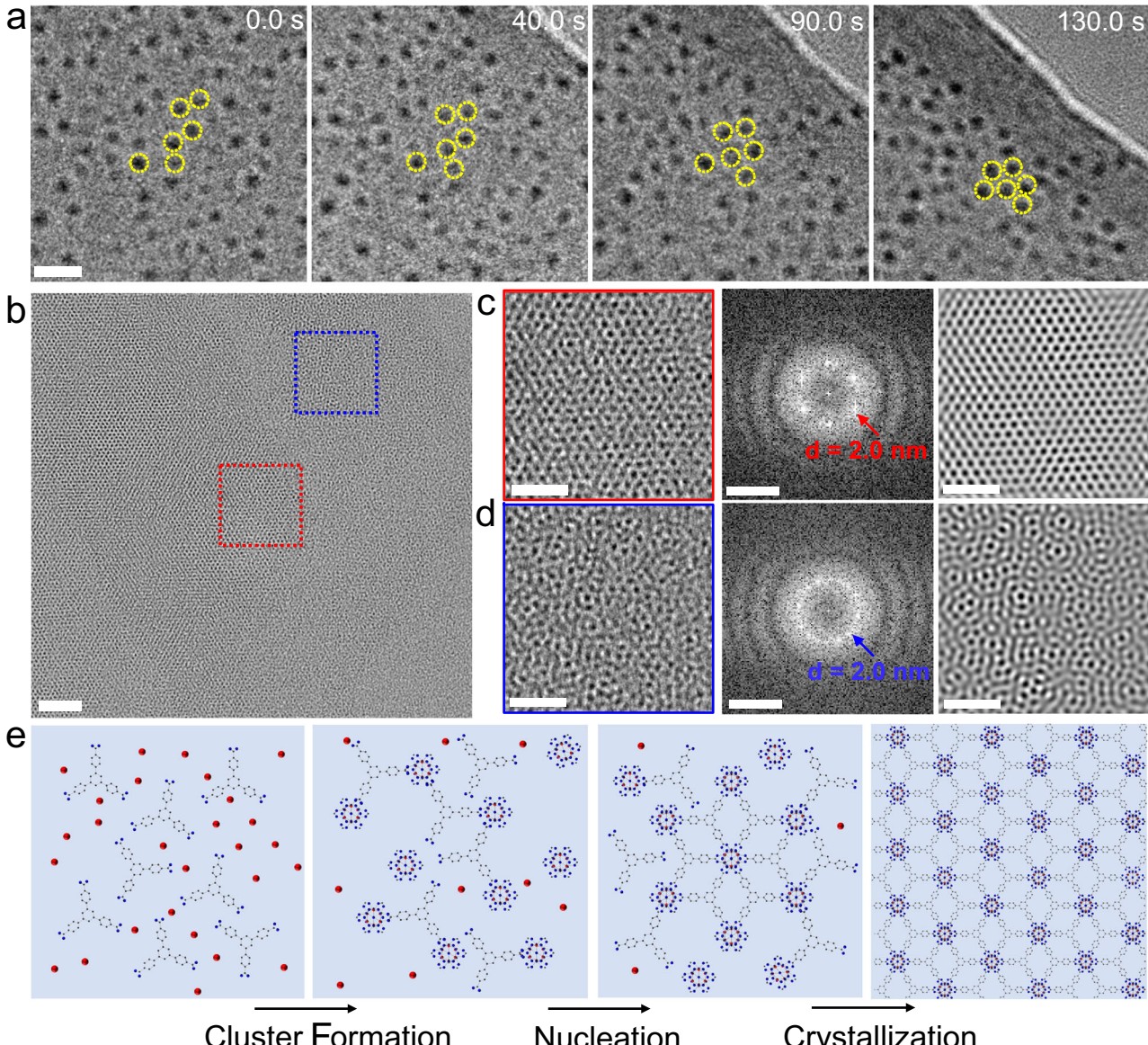

**Fig. 2 | The formation process of Hf-MOLs. a** Snapshots from Supplementary Movie 3 showing the Hf-clusters assembled into a hexagonal arrangement. **b** HRTEM image of Hf-MOLs after baking the growth solution at 120 °C for 30 min (early-stage product). **c** Enlarged image (left panel) of the red square area shown in **b**, showing the bulk crystalline structure in more detail; the corresponding FFT (middle panel) and IFFT (right panel) images of the red region are also displayed. **d** Enlarged images (left panel) of the blue square region shown in b, showing the disordered structure of the MOL; the corresponding FFT (middle panel) and IFFT (right panel) images of the blue region are displayed. **e** Schematic of MOLs formation process depicting the formation of clusters, followed by the aggregation and spontaneous nucleation and growth to form a periodic structure. The red, blue and black atoms represent Hf, O and C, respectively. Scale bars: **a** 10 nm; **b** 20 nm. Scale bars of TEM (left panel) and IFFT (right panel) images in **c** and **d** are 10 nm. Scale bars of FFT (middle panel) images in **c** and **d** are 1 1/nm.

formed in the early stage. Our results show that clusters first form in solution and are then complexed with ligands to form paracrystalline solids, followed by the arrangement of the cluster-ligand complex into a crystalline film (Fig. 2e).

## Identification of local structures of Hf-MOLs using HAADF-STEM

With the aim of gaining local structural information of Hf-MOLs, spherical aberration ($C_s$) corrected HAADF-STEM characterization was conducted on the Hf-MOLs. The Hf-clusters create high contrast for STEM characterization, which is helpful to observe local defect structures in the MOL. Figure 3a displays some typical missing-cluster regions. The inset shows the enlarged region of the yellow square, where one cluster was missing from the center, indicated by the white arrow. The structural model of the point defect region is shown in Supplementary Fig. 9. In addition to the one cluster missing defect, a crack was observed

within the bulk Hf-MOLs, where a row of clusters is lacking, as shown in Fig. 3b. Furthermore, we investigate the stacking behaviors of Hf-MOLs. Because of the random stacking of the bottom and top layers in some local areas, some disordered structure is created (marked with the white frame), as shown in Fig. 3c. We also observe an interface formed due to different layer stacking of Hf-MOL nanosheets, as displayed in Fig. 3e, f. There are many disordered regions formed in the interface, as indicated by the yellow square in Fig. 3e. The top part shows interpenetrated layers, which are caused by the interlocked stack packing of different layers of Hf-MOLs (Supplementary Fig. 10). However, the bottom part maintains its hexagonal structure because of fully coincident stack packing (Fig. 3f and Supplementary Fig. 10).

We also observed edge termination structures of the MOL using HAADF-STEM. After observing various types of crystalline sheet edges present in the samples, we found two representative types of surface

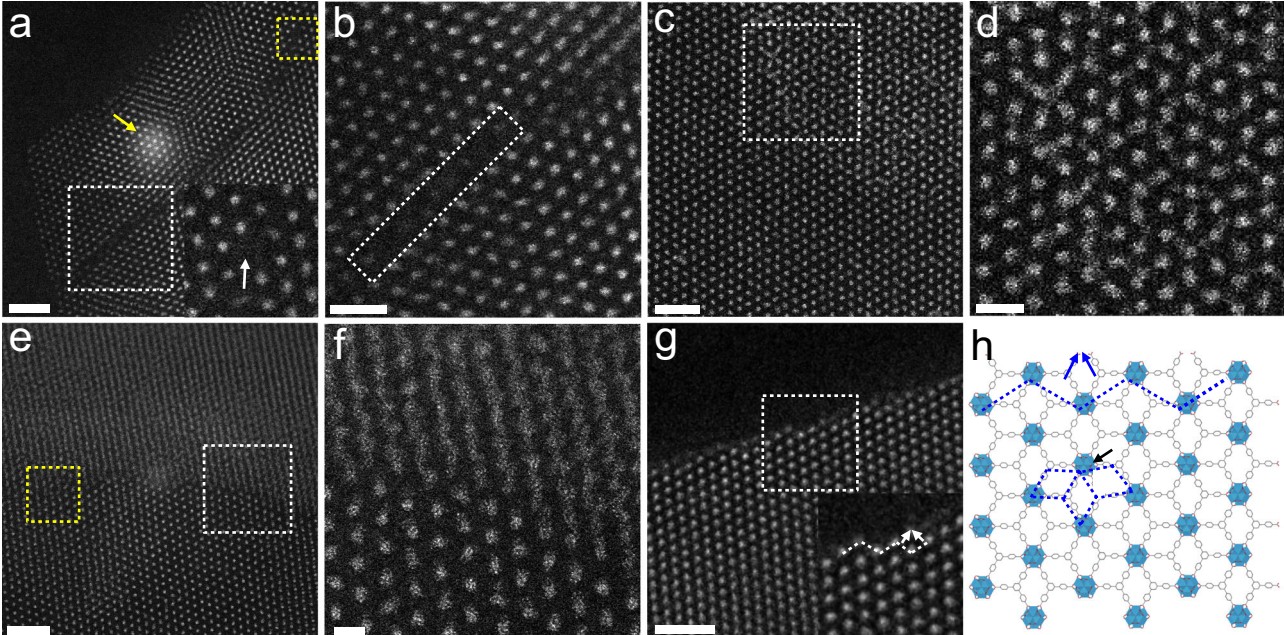

**Fig. 3 | Revealing local structures of Hf-MOLs using HAADF-STEM. a** HAADF-STEM image of MOLs shows the missing-cluster regions. The inset is the enlarged region of the yellow frame, which shows one cluster missing from the center, as indicated by the white arrow. The area indicated by the yellow arrow was caused by the electron beam irradiation for >10 s. **b** Enlarged region of the white frame in **a**, where a row of clusters disappeared. **c** HAADF-STEM image of MOLs shows connected clusters and local disorder areas within the bulk of Hf-MOLs. Different layers of Hf-MOLs stack to form the yellow polygonal area, and some disorder structures (yellow square) are generated at the interface. **d** A zoomed-in area of the white square region in **c**. **e** HAADF-STEM image shows the boundary between Hf-MOLs. **f** Enlarged region of the white frame in **d**. **g** HAADF-STEM image of Hf-MOLs shows the step-edge site surface structure. The inset shows the enlarged region of the white square, and the white arrows indicate the position to add a new cluster. **h** Schematic illustration of the loop structure contributes to epitaxial growth of MOLs. The blue arrows show the connection site to add a new cluster, where the new cluster could form three rings (similar to the position indicated by the black arrow) with neighboring clusters. Scale bars: **a**, **c**, **e**, **g** 10 nm; **b** 5 nm; **d**, **f** 2 nm.

terminations in the Hf-MOLs (Supplementary Fig. 11). One of them exhibited a loop surface, as shown in Fig. 3g, while another terminated with a flat surface. The inset shows the enlarged region of the white square, where step-edge sites, another commonly observed edge structure, are also present.

### Electron ptychography of Hf-MOLs to image the ligand

HAADF-STEM can produce high-quality images of Hf-MOLs at the cluster length scale. However, it is not sensitive to light elements in the sample, limiting our ability to observe the structure of ligands within Hf-MOLs. The ultrathin nature of MOLs makes them ideal samples for high-resolution characterization methods with low doses using electron ptychography[48–51]. This technique provides a readily interpretable phase imaging and great sensitivity for imaging light elements at atomic resolution[52]. To fully understand the structure of Hf-MOLs, electron ptychography was performed. Electron ptychographic reconstruction of Hf-MOLs was conducted using the 4D Camera, an in-house developed direct electron detector with $576 \times 576$ pixels and a frame rate of 87 kHz (Fig. 4a). Figure 4b shows a representative electron ptychographic phase reconstruction image that was acquired at an accelerating voltage of 80 keV. The high-resolution phase image enables resolving individual Hf clusters. Some Hf atoms even could be identified very clearly from the contrast of the enlarged image. An FFT of the reconstruction shows a transfer of information of as high as $1/2.36\ \text{Å}^{-1}$ (Fig. 4c). The surface structure was also imaged by electron ptychography, as shown in Fig. 4d. A uniform nanoscale ligand layer was observed at the surface of Hf-MOLs, as indicated by the white dash line in Fig. 4d (Supplementary Fig. 12). To enhance the signal-to-noise ratio and thus improve the level of information obtained from the phase image, we performed 2D class averaging of 70 unit cells of the reconstructed phase image using the cryo-EM software package RELION[53]. In this process, individual clusters are selected and aligned to each other

and then classified based on similar features, yielding a single low-noise class average depicted in Fig. 4e. In this way, Hf-clusters and connector ligands were simultaneously observed. We can see that there are six ligands around each cluster, and that each ligand correlates with three clusters to form the 2D network. To further confirm the observed structure are consistent with the existing theoretical model, we performed ptychographic reconstruction of a simulated dataset of Hf-MOLs with four layers, as shown in Fig. 4f. The simulated results show the same characteristics as the reconstructed phase image.

## Discussion

We studied the formation and local features of ultrathin MOLs with tolerable dose rates via combined TEM techniques. Compared with most reports that only use one single microscope technique, the use of combined TEM techniques provides a complementary understanding of the growth and structure of MOLs. Through in-situ observation and the characterization of ex-situ intermediate products, we infer that the structural evolution of Hf-MOLs can be divided into three stages, as illustrated in Fig. 2e. Initially, an incubation time is required for the formation of clusters, corresponding to the initial formation of clusters. The clusters appear in the solution before the formation of Hf-MOLs. During the second stage, generated clusters correlated with neighboring ligand to form dimers, trimers or multimers, resulting in a disordered structure. Therefore, non-crystalline solid precursors form in the solution in this process. In some local areas with high concentration of clusters, a hexagonal close packing structure will appear first. This is the initial nucleation of the Hf-MOLs, and external stimuli such as heating and stirring will speed up this process. In the following process, clusters will approach and adhere to the nuclei, eventually forming large 2D nanosheets.

High-resolution STEM images enable us to determine various local structural defects within MOLs. We note that many regions exist with

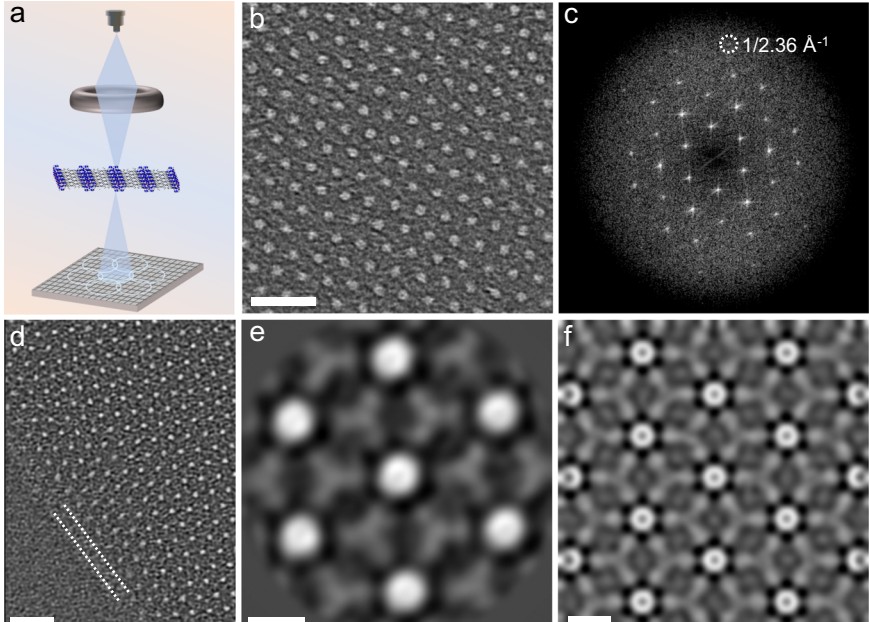

**Fig. 4 | Electron ptychographic reconstruction of Hf-MOLs using a high frame rate direct electron detector. a** Experimental setup for electron ptychography. **b** Ptychographic reconstruction of MOLs from 4D-STEM data. **c** Intensity of Fourier transform of the reconstructed phase image of Hf-MOLs, in which the dashed white circle represents information transfer to 1/2.36 Å$^{-1}$. **d** Ptychographic reconstructed phase image of the surface/interface of the MOLs. The white dash line shows the nanoscale surface ligand layer. **e** 2D class average of 70 unit cells of the reconstructed phase image. The image shows Hf-clusters and BTB ligands. **f** Ptychographic reconstruction of a simulated dataset of Hf-MOLs with four layers. Scaler bars: **b, d** 5 nm; **e, f** 1 nm.

missing clusters and disorder structures. Clusters missing may enhance the accessibility of active sites, and the defects serve as the catalytically active Lewis acid sites for chemical reactions, thereby improving catalytic activities for some reactions[20]. In addition to defect regions, two representative types of surface terminations were observed, namely the loop surface and the flat surface. We propose that the edge-sites play important roles in adding clusters to the surface. The loop structure creates an open site (indicated by the blue arrow in Fig. 3h) to add a new cluster, where the new cluster binds with three adjacent clusters and thus increases the stability of the surface clusters via bridge ligands. This is similar to the position marked by the black arrow in Fig. 3h. Our observation is consistent with the previous report that it is thermodynamic preferable for MOFs to grow at step-edge sites rather than at flat surface sites[32].

These high-resolution results confirm that Hf-MOLs are ideal samples for investigating the local structures of organic framework materials using HAADF-STEM, including defects, stacking, surface and interface structure. Moreover, the ultrathin nature of MOLs is suitable for high-resolution characterization using electron ptychography. Therefore, we have obtained ptychographic reconstructed phase images of the connector ligands with a characteristic three-pointed star structure. Although we use Hf-MOLs with a known crystal structure for imaging in this work, we believe electron ptychography is an excellent technique to reveal the structure of new low-dimensional hybrid materials.

In summary, we have performed a systematic study of the growth mechanism and local structures of Hf-MOLs using in-situ TEM, Cryo-EM, HAADF-STEM and electron ptychography. We revealed the growth process of MOLs in solution at cluster scale. The TEM results demonstrate that Hf-clusters are first generated in the solution and coordinated with ligands to form non-crystalline precursors, followed by the self-assembly process of the Hf-clusters to form hexagonal arranged nuclei from rich clusters regions. With the high-resolution capability of characterizing Hf-MOLs via combined TEM techniques, we achieved the real-space observation of structural defects and surface/interface structure of MOLs with sub-unit cell resolution. We reported important

local features of MOLs, including missing clusters, disorder structure, loop and flat surface terminations. Furthermore, the high-resolution of ptychographic reconstruction allows us to identify the organic linkers. Our observations on the formation and local structures provide insights on improving the MOLs/MOFs design, thus leading to performance enhancement of the functionalities of application in another research field, such as catalysis.

## Methods

### Materials
All commercially available chemicals including Hafnium (IV) chloride (HfCl$_4$, 99.9%, Strem), 1,3,5-Tris(4-carboxyphenyl)benzene (H$_3$BTB, 97%, Jinan Henghua), Formic acid (HCOOH, 99%, Amethyst), N,N-Dimethylformamide (DMF, 99.9%, J&K Scientific) were used without further purification unless otherwise indicated. Ultrathin carbon film on lacey carbon support film for high-resolution characterization was purchased from Ted Pella.

### Metal-organic layer synthesis
The synthetic conditions for Hf-MOLs were optimized as previously reported[40]. Briefly, 14 mg HfCl$_4$, 12.5 mg H$_3$BTB were dissolved in a mixture solvent of 0.8 mL HCOOH, 2.0 mL Dimethylformamide (DMF) and 0.3 mL H$_2$O in a Pyrex vial. The solution was sonicated for 10 min and then heated up to 120 °C for 1 day unless otherwise specified. Then, the MOLs suspension was centrifuged and washed three times using DMF to obtain a white product. The early-stage products were synthesized by heating the growth solution at 120 °C for 30 min. To synthesize the 3D Hf-MOFs, 1.35 mL HCOOH and 0.1 mL H$_2$O were used.

### Liquid cell fabrication and TEM characterizations
A homemade enclosed liquid cell was used for the in-situ observation. Liquid cell was fabricated using the same procedures as described in the previous report[54]. An ultrathin (~10 nm) silicon nitride membrane was used to improve the spatial resolution of the liquid cell. A droplet of growth solution was filled into the liquid reservoir and then sealed

with glue for TEM imaging. A Tecnai F20 TEM operated at 200 keV and ThemIS (Thermo Fisher Scientific) operated at 300 keV were used for in-situ imaging.

We perform HRTEM, HAADF-STEM, EDS for Hf-MOLs using an aberration-corrected electron microscope (TEAM 1 and ThemIS) operated at 80–300 keV at the National Center for Electron microscopy of Lawrence Berkeley National Laboratory and Thermo Fisher Scientific. The sample characterization at cryogenic temperature was achieved using a Gatan 636 cryo-holder.

### Electron ptychography

TEM experiments were performed on an aberration-corrected electron microscope (TEAM 0.5) with a probe-forming aperture semi-angle of 11.4 mrad at an electron energy of 80 keV. The four-dimensional scanning transmission electron microscopy (4D-STEM) dataset from Hf-MOLs crystals was acquired using the 4D Camera[55], an in-house developed direct electron detector with $576 \times 576$ pixels and a frame rate of 87 kHz. To achieve a low total electron dose, the sample was brought close to focus at a resolution of 57 kx by focusing on the low-resolution Moire lattice of the sample. Then, the beam was blanked, and the resolution increased to 1.3 MX, corresponding to the pixel size of 0.876 Å. We took scans with $512 \times 512$ positions at this step size with a scanning area of $45.6 \times 45.6 \, nm^2$, a fluence of $864 \, e^- \cdot Å^{-2}$, corresponding to an average electron flux of $0.287 \, e^- \cdot Å^{-2} \cdot ms^{-1}$.

### Reconstruction algorithms

Raw 4D-STEM data was electron counted with the open-source python software stempy and converted to a sparse linear-index encoded electron event representation (EER)[56]. Preprocessing like centering and binning of the diffraction patterns was performed directly on the sparse EER format. Phase images were then reconstructed using a real-time GPU implementation of the single-sideband ptychography method[57,58]. The defocus was determined by a grid search with the phase standard deviation as a quality criterion.

### Class averaging

2D class averaging was performed with the RELION software for cryo-EM applications[59]. First, 70 cluster centers were picked manually and then extracted with a crop window of $6.33 \, nm \times 6.33 \, nm$. 2D class averaging with ten candidate classes was then performed on these 70 cropped clusters, yielding a single low-noise class depicted in Fig. 4e.

### 4D-STEM simulation and TEM simulation

4D-STEM datasets of the Hf-MOL in Fig. 4f were simulated with the same parameters as the experiment with the abTEM simulation package[60], using the PRISM algorithm[61]. TEM simulation was performed by a custom-written Python script based on the codes provided in the book Advanced Computing in Electron Microscopy[47].

### Electron beam effects

In-situ imaging of clusters formation and self-assembly in DMF solution was performed at 200 keV. It is important to mention that we didn't see obvious structure damage or dissolution during the whole imaging process, which indicates that these clusters are relatively stable in solution. We evaluated the stabilities of Hf-MOLs under a 300 keV accelerated electron beam at low temperature. Cooling the sample with liquid nitrogen contributes to reducing the thermal effect and slow diffusion of ionizing species. We found that MOLs begin to lose crystallinity when the dose rate goes up to $3200 \, e^- \cdot Å^{-2} \cdot s^{-1}$ at low temperature in TEM mode (Supplementary Fig. 13). This value sets the upper limits of electron dose for image acquisition. The stability test under electron beam irradiation indicates that the Hf-MOL structure can be well maintained at an electron dose rate of $1750 \, e^- \cdot Å^{-2} \cdot s^{-1}$ for at least 49.5 s at room temperature (Supplementary Movie 4). A long-time electron beam exposure experiment was performed to understand the damage

process of Hf-MOL at an electron dose rate of $1750 \, e^- \cdot Å^{-2} \cdot s^{-1}$ (Supplementary Fig. 14). We found that the Hf-clusters get connected and form an amorphous region under the beam irradiation. For the HAADF-STEM measurements, control experiments were performed to exclude the influence of the electron beam. The results show that MOLs will not lose their crystallinity when the dose is $<106 \, e^- \cdot Å^{-2}$ at room temperature in STEM mode (Supplementary Fig. 15). Therefore, we could achieve HRTEM/STEM of MOLs at cluster scale (~1 nm) at both room temperature and low temperature without breaking the periodic structure. To achieve a low total electron dose, the sample was brought close to focus at low magnification. Then, the beam was blanked, and the magnification was increased to high magnification for image capture for HAADF-STEM and electron ptychography. During the imaging, the electron beam only influences the square area scanned. Therefore, we could also focus the sample on a certain place and acquire images in a nearby position that remains unexposed. In many cases, the beam is blanked while moving around the sample to reduce beam exposure time, and only unblanked when acquiring images. We observe that for the high dose rate ($800 \, e^- \cdot Å^{-2}$) ptychography reconstruction, the diffractional signal reaches an overall higher resolution (Supplementary Fig. 16). The three-pointed star morphology of ligand structure (Fig. 4e) and well-maintained hexagonal arrangement of clusters indicate that the dose of electron for imaging is sufficiently low to avoid structural damage and preserve the integrity of the original sample.

## Data availability

All TEM data supporting the findings of this study are contained in the paper and its Supplementary Information files. Supplementary Movie 1-3 contain in-situ liquid phase TEM data showing cluster formation, movement of clusters, and self-assembly of clusters. Supplementary Movie 4 shows the stability of Hf-MOLs under electron-beam irradiation at room temperature in TEM mode. All other relevant data are available from the corresponding author (Mary C. Scott) on request.

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

## Acknowledgements

Work at the Molecular Foundry was supported by the Office of Science, Office of Basic Energy Sciences, of the U.S. Department of Energy under Contract No. DE-AC02-05CH11231. P.M.P. and M.C.S. are supported by the Strobe STC research center, Grant No. DMR 1548924. Y.Z. was supported by the Samsung Advanced Institute of Technology. H.Z. acknowledges the support of the U.S. Department of Energy, Office of Science, Office of Basic Energy Sciences, Materials Sciences and Engineering Division under Contract No. DE-AC02-05-CH11231 within the KC22ZH program. We gratefully thank Dr. Peter Ericus and Dr. Hao Yang for their helpful discussion.

## Author contributions

X.P., P.M.P. and Q.Z. conceived and designed the experiments. X.P., Q.Z. and H.Z. designed and conducted the in-situ liquid phase TEM experiment. P.M.P. performed the electron ptychography experiment. X.P., P.C., L.C. and Y.Z. performed the materials synthesis and ex-situ characterization. X.P. wrote the manuscript with inputs from all the other authors. M.C.S., H.-G.L., S.-G.S. and C.W. supervised the research.

## Competing interests

The authors declare no competing interests.
