## [Peer Review File · Nature Communications]

Observation of Formation and Local Structures of Metal-Organic Layers via Complementary Electron Microscopy TechniquesREVIEWER COMMENTS

Reviewer #1 (Remarks to the Author):

In this manuscript, X. Peng, et al., used several complementary electron microscopy techniques to study the growth processes and local structures of metal-organic layers. There are at least three important findings. First, the multistep formation process is clearly illustrated using in-situ liquid-cell TEM. Second, multiple defect-structures and two types of surface terminations are observed. Third, the organic ligands are directly imaged, although after averaging over many unit-cells. These are all problems with general interests in related fields. However, due to beam damage issues in TEM, such structural information is extremely challenging to obtain. Therefore, this manuscript does have the potential to be appear in Nat. Commun.. However, the connection between structure and properties is weak and sometimes handwavy. Detailed reasons about this assessment together with additional crucial questions are listed below:

1. Two types of surface terminations are observed, and the authors predicted the edge-sites are the preferred places for adding new clusters during growth. If it is thermodynamic preferable to grow at step-edge sites, why are there so many flat surfaces? From supplementary Fig. 9, it is not statistically different in the density of the two types of surface terminations. The authors should provide more in-depth discussions or may perform some simulations.

2. They vaguely claimed that missing clusters and disorder structures are closely related to the stacking behaviors. It is clear that missing clusters are not necessarily related to the stackings, e.g., Fig. 3a. It is natural to say active sites from the missing clusters and surface terminations may improve catalytic activities but not obvious why stacking caused disorder enhances the chemical reactions. Stacking caused disorder may not be so active. The authors should be more careful with such claims. In addition, no evidence has been provided to show controlling stacking pathways of Hf-MOLs is the most useful way to control the defective structures.

3. The authors only show that electron dose less than 10^6 e/A² does not cause visible damage when using 300 kv beam energy in STEM at room temperature. But they used 864 e/A² dose (more than 8 times) and 80 kv beam energy for ptychography. They should show what occurs in such imaging conditions. It is well known that lower beam energy for insulators suffers larger ionization damage. Obvious damage can be indicated from the intensity variations (or artifacts) in Fig. 4b and 4d. The contrast from organic linkers in the class averaged results in Fig. 4e is much lower than that from simulated results in Fig. 4f, which is also evidenced as the damage effect. Furthermore, the surface structure claimed from Fig. 4d can be induced by the electron beam. The authors should clearly clarify all these problems.

4. The authors should not pretend that they are the first who knows/shows the advantages of electron ptychography. It has already been demonstrated that ptychography is uniquely sensitive to both heavy and light atoms simultaneously, e.g., H. Yang, et al., *Ultramicroscopy* 180, 173 (2017). Low dose capabilities have been demonstrated with doses lower than 500 e/A² at resolutions better than 2 Å, e.g., J. Song, et al., *Sci. Rep.* 9, 3919 (2019) and Z. Chen, et al., *Nat. Commun.* 11, 2994 (2020). Images from both previous works have much better quality than those shown in Fig. 4 in this manuscript. There is no reason why the authors did not cite any of these earlier works. It is really misleading for audiences who are not experts in electron ptychography (which is a growing but still small community), especially in a paper that claims the advantages of new imaging techniques but maybe targets more on readers from material sciences.

Minor points:

1. Fig. 3a-d labels are missing.
2. The characteristic distance of 2 nm can be more clearly illustrated by pointing out the spatial frequency of 2 nm in the FFT in Fig. 2c and 2d. In addition, a scale bar in the FFTs may be needed.
3. What is the dose rate for the 14 s illumination in Supplementary Fig. 13 in STEM-HAADF? It is

confusing about the total dose tolerance in STEM at room temperature. If more doses were imposed between two acquisitions shown as in Supplementary Fig. 13, then the dose tolerance should be much higher than $106 \text{ e}/\text{Å}^2$.

4. In the method part, it is stated that 287 cluster centers were used for the class averaging. But in the main text, it is stated that 254-unit cells were used. Is there any typo or the cluster centers are not the same as unit cells?

Reviewer #2 (Remarks to the Author):

Overall:

This is an interesting TEM-based study of the 2D MOF, specifically Hf-MOL. The ways/methods that authors used to gain insight into the Hf-MOL formation are good. But the conclusions they are making with the results/observations in-hand are not properly supported. More results and clarifications are needed before this manuscript would be acceptable for publications.

Fundamental questions that need justification:

1. It is not clear that the in-situ TEM observations that authors presented in this work is the natural formation of the Hf-MOL without non-negligible role from e-beam of TEM. More results are needed to eliminate possible influential role of e-beam of TEM.

Specific questions that need clarification:

1. For the results based on Sup. Figure 4 and Movie 1: When liquid is injected into the chip, due to huge mass difference between the chip and liquid, I would suspect that liquid will rapidly cool down. Do authors know the temperature of the liquid when it was observed in the TEM? Also, how long did it take from "injection" to "imaging" it in TEM? Since it takes some time, due to injection, vacuum check and finding window/focusing in the TEM (5-10 min perhaps), wouldn't authors expect that formation of the Hf clusters occur before that 3.5 min video was taken. Please provide clarifications about Hf cluster formation kinetics in this context.

2. The movie S2 looks like TEM sample drift. Can authors show that it indeed shows clusters moving and not TEM sample drift. If it is indeed Hf cluster moving, why all clusters are moving in same direction synchronous? Wouldn't they move randomly? Clarification is needed.

3. In the discussion to Fig. 2b-d, authors show that crystal core formed at the "center" and disorder is at the edges/surface. The "center" of view is the region where there is more effect from e-beam of TEM. Can authors prove that, what they see, is not an effect of TEM e-beam, but natural Hf-MOL formation. I would guess that in the liquid, there is no "center", and all Hf-MOL formations should happen randomly in all places.

4. There results authors presented, doesn't support the claim (on line 153): "Our results show that clusters first form in solution and are then complexed with ligands to form amorphous solids, followed by the arrangement of the cluster-ligand complex into a crystalline film (Fig. 2e)." The results don't show/say anything about ligands. It is not even clear if there are ligands there. At this point this is only speculation, which might not be true.

5. Line defects in 3D or 2D crystals has district definition. The observation in Fig 3a-b, doesn't constitute line defect and needs to be explained better.

6. You cannot have "amorphous striped morphology" by definition. If there are "strips", it is not amorphous. Better description is needed.

Technical

1. In the abstract, it reads "...clusters, point and line defects, dislocations, loop and flat surface terminations, and ...". Dislocation is a line defect. Remove "dislocation" or "line defect".

2. There is a recent paper in Chem Mater (S. Ghosh et al (2021)) that has comprehensive discussion on e-beam damage of MOF in TEM.
3. In Supplementary figure 1, could authors re-check their files? Low-mag "120 min" image in panel (b) appears to have different magnification (based on size of wrinkled edges), even the scale bars are the same size. Also TEM operational conditions and e-beam doses should be indicated here (for panel b and c) to be sure there is no beam-damage issues here.
4. Figure 1(a) need labeling of the atoms. In the models, which are Hf, C and O atoms? The models also need to be presented from other crystallographic orientations to have full picture of the structure.
5. Proper (earlier) reference to Z-contrast ADF-STEM are: S.J. Pennycook, L.A. Boatner, Nature 336 (1988) 565 and S.J. Pennycook, Ultramicroscopy 30 (1989) 58., and not the reference [41] cited in this paper.
6. The sentence (on line 123) "Typically, Hf-MOL formation takes place under solvothermal conditions at temperature." is not complete.
7. In the caption to Fig. 2, "movie S1" should be replaced with "movie S3".
8. In Fig. 3, labels a-d are missing.

Reviewer #3 (Remarks to the Author):

The work described in the manuscript entitled "Observation of Formation and Local Structures of Metal-Organic Layers via Advanced Electron Microscopy" is significant for the rapidly developing cross-field of electron microscopy for the beam-sensitive materials investigations. Together with this, it gives a unique insight on the formation mechanisms of metal-organic materials. To the knowledge of the reviewer, there are no similar works pushed at the moment. I would recommend this manuscript for publication after the minor revision.

1. In the past 5-7 years the work on the TEM for the metal-organic materials was very intense and I would recommend adding the references to more papers published by now - for example, on TEM, on STEM, on iDPC. It could give a better understanding for the reader about the current field state and the place of this work.
2. In general in the text the superior words, such as unique, for example, are a bit overused. I would recommend aligning the manuscript with the more accurate wording.
3. Adding simulated images and overlapping the structure models to the experimentally recorded images would significantly improve the quality of work and the discussion.

Reviewer: 1

In this manuscript, X. Peng, et al., used several complementary electron microscopy techniques to study the growth processes and local structures of metal-organic layers. There are at least three important findings. First, the multistep formation process is clearly illustrated using in-situ liquid-cell TEM. Second, multiple defect-structures and two types of surface terminations are observed. Third, the organic ligands are directly imaged, although after averaging over many unit-cells. These are all problems with general interests in related fields. However, due to beam damage issues in TEM, such structural information is extremely challenging to obtain. Therefore, this manuscript does have the potential to be appear in Nature Communications. However, the connection between structure and properties is weak and sometimes handwavy. Detailed reasons about this assessment together with additional crucial questions are listed below:

1. Two types of surface terminations are observed, and the authors predicted the edge-sites are the preferred places for adding new clusters during growth. If it is thermodynamic preferable to grow at step-edge sites, why are there so many flat surfaces? From supplementary Fig. 9 (Fig. 11), it is not statistically different in the density of the two types of surface terminations.

The authors should provide more in-depth discussions or may perform some simulations.

[Re] We thank the referee's comment. When two secondary building units (SBUs) on the MOF are linked by one ligand to form a "loop" structure, two molecules of formates are released, which increases the number of freely moving entities in solution and provides an entropy gain¹. Therefore, we consider that the entropy gain can contribute to the thermodynamically preferential growth at step-edge sites. The appearance of the flat surfaces does not conflict with adding new clusters at step-edge sites. As shown in Figure R1, new clusters are added to the step-edge sites to eventually form a flat surface. In addition, similar results have been reported the surface of some representative MOFs (e.g., ZIF-8) are not atomically pristine. The co-existence of step-edge sites and flat surfaces are clearly observed from the high-resolution TEM images using cryo-electron microscopy².

Figure R1. Schematic illustration of the epitaxial growth of MOFs from the step-edge sites. The blue spheres represent newly added clusters, and the blue square region indicates a flat surface.

2. They vaguely claimed that missing clusters and disorder structures are closely related to the stacking behaviors. It is clear that missing clusters are not necessarily related to the stackings, e.g., Fig. 3a. It is natural to say active sites from the missing clusters and surface terminations may improve catalytic activities but not obvious why stacking caused disorder enhances the chemical reactions. Stacking caused disorder may not be so active. The authors should be more careful with such claims. In addition, no evidence has been provided to show controlling stacking pathways of Hf-MOFs is the most useful way to control the defective structures.

[Re] Thanks for the comment. After reviewing our results, we agree with the reviewer's comment that the descriptions of "missing cluster formation is closely related to the stackings" and "stacking caused disorder enhances the chemical reactions" are not appropriate. Following this comment, we have removed the related description in the revised manuscript.

3. The authors only show that electron dose less than $106 \text{ e}/\text{A}^2$ does not cause visible damage when using 300 kv beam energy in STEM at room temperature. But they used $864 \text{ e}/\text{A}^2$ dose (more than 8 times) and 80 kv beam energy for ptychography. They should show what occurs in such imaging conditions. It is well known that lower beam energy for insulators suffers larger ionization damage.

[Re] Thank you for raising these important questions. We use 80 keV beam energy in our study for three reasons:

- 1) The direct electron detector we used for the measurements, a prototype of the 4D camera

monolithic active pixel detector, is currently optimized for 80 keV electrons, and has 3x reduced efficiency at 300 keV.

2) our sample, a multi-layer Hf-MOL on ultrathin carbon substrate, is less than 20 nm thick. For imaging the organic carbon linkers of this thickness, the energy that extracts the most information per damage is below 100 keV.³ Previous studies mostly imaged thicker regions of bulk MOF crystals, where the optimal energy for imaging is higher.

3) The high frame rate of our detector allows us to image the sample in a high dose-rate regime, where we have observed less damage to the structure than imaging with low dose rate. Figure R2 below shows the radial integral of the power spectrum of reconstructions from adjacent regions, collected with different dose rates. The reconstruction obtained with a single, high dose-rate exposure at $800 \text{ e}^- \cdot \text{\AA}^{-2}$ produces a higher diffraction signal than the sum four reconstructions with data from a close-by area with the same thickness obtained with $200 \text{ e}^- \cdot \text{\AA}^{-2}$ exposure. This indicates that the crystalline structure of the Hf-MOL is well preserved with our measurement protocol, and that resolution is improved compared to a low-dose rate protocol. A recent study on a different MOF has shown that loss of crystallinity happens at the early stage of radiation damage.⁴ Diffraction signal to high resolution indicates that the atomic structure is preserved with our measurement protocol.

Figure R2. A single high-dose-rate exposure produces higher diffraction signal than many low-dose-rate exposures

Obvious damage can be indicated from the intensity variations (or artifacts) in Fig. 4b and 4d. The contrast from organic linkers in the class averaged results in Fig. 4e is much lower than that from simulated results in Fig. 4f, which is also evidenced as the damage effect. Furthermore, the surface structure claimed from Fig. 4d can be induced by the electron beam.

[Re] We found that the maximum standard deviation criterion we used to determine the reconstruction defocus does not estimate the true defocus, which leads to an overall blurred unit cell average reconstruction and correlated noise in the single-side band ptychography image. We have adjusted the defocus of the reconstruction now by careful comparison with simulation and now achieve a better match with the simulated reconstruction.

We have improved our simulations by including a randomized amorphous carbon substrate, which was omitted in previous reconstructions from simulated data, 10 frozen phonons, and the defocus spread and chromatic aberrations of the TEAM 0.5 microscope in the simulations with the open source abTEM simulation code. These improvements in the simulation lead to overall better agreement with the experimental reconstruction.

We also performed in-situ observation of structural evolution under electron beam irradiation to compare with the results in Fig. 4d. As shown in movie 4, there is no obvious damage to the MOL structure for more than 49.5 s at a high dose rate of $1750 \text{ e}^- \cdot \text{\AA}^{-2} \cdot \text{s}^{-1}$ under TEM mode at room temperature. In addition, if the surface layer was formed due to the damage of surface clusters, we should be able to observe it under HAADF-STEM mode because Hf-clusters can still provide high contrast even as they connect with neighboring clusters during beam damage. In our system, we could even get a high-resolution STEM image at a dose rate of $16984 \text{ e}^- \cdot \text{\AA}^{-2}$ by unblanking the beam only when acquiring an image, the regions nearby will remain unexposed. No surface layer was observed from the high-resolution STEM image (Figure R3). We use the same method for the data acquisition in Fig. 4d, and the dose rate is much smaller than this. So, it is clear that the surface layer was not caused by the beam effect.

Figure R3. High-resolution HAADF-STEM images of Hf-MOL at a dose rate of $16984 \text{ e}^- \cdot \text{Å}^{-2}$.

Scale bar 2 nm.

4. The authors should not pretend that they are the first who knows/shows the advantages of electron ptychography. It has already been demonstrated that ptychography is uniquely sensitive to both heavy and light atoms simultaneously, e.g., H. Yang, et al., *Ultramicroscopy* 180, 173 (2017). Low dose capabilities have been demonstrated with doses lower than $500 \text{ e}/\text{Å}^2$ at resolutions better than 2 Å, e.g., J. Song, et al., *Sci. Rep.* 9, 3919 (2019) and Z. Chen, et al., *Nat. Commun.* 11, 2994 (2020). Images from both previous works have much better quality than those shown in Fig. 4 in this manuscript. There is no reason why the authors did not cite any of these earlier works. It is really misleading for audiences who are not experts in electron ptychography (which is a growing but still small community), especially in a paper that claims the advantages of new imaging techniques but maybe targets more on readers from material sciences.

[Re] Thanks for pointing this out. We highly value researchers in the electron ptychography community who develop this technique to achieve high-resolution characterization of beam-sensitive materials. Following the referee's suggestion, some important earlier works of electron ptychography are cited in the revised manuscript. We have added more discussion on previous work of electron ptychography on materials characterization in the revised manuscript. (Highlighted on page 9).

Minor points:

1. Fig. 3a-d labels are missing.

[Re] We have added the labels a-d in the new Fig. 3

2. The characteristic distance of 2 nm can be more clearly illustrated by pointing out the spatial frequency of 2 nm in the FFT in Fig. 2c and 2d. In addition, a scale bar in the FFTs may be needed.

[Re] We thank the referee's comment and suggestion. We agree that the spacing of 2 nm between SBUs can be more clearly presented from the FFT image in Fig. 2c and 2d. Following the suggestion, we mark the 2 nm distance in the new Fig. 2c and 2d. Scale bars of FFT images are also added in Fig. 2c, d.

3. What is the dose rate for the 14 s illumination in Supplementary Fig. 13 in STEM-HAADF? It is confusing about the total dose tolerance in STEM at room temperature. If more doses were imposed between two acquisitions shown as in Supplementary Fig. 13, then the dose tolerance should be much higher than $106 \text{ e}/\text{Å}^2$.

[Re] The illumination dose rate is the same as the STEM imaging condition, namely a fluence of $106 \text{ e} \cdot \text{Å}^{-2}$. We agree that the dose tolerance is much higher than $106 \text{ e} \cdot \text{Å}^{-2}$, which can also be seen from our reply to question 3 above.

4. In the method part, it is stated that 70 cluster centers were used for the class averaging. But in the main text, it is stated that 254-unit cells were used. Is there any typo or the cluster centers are not the same as unit cells?

[Re] Thanks for pointing this out. There is no difference between the cluster and unit cells. We use 70 unit cells for the class averaging. We have corrected this in the revised manuscript.

Reviewer #2 (Remarks to the Author):

Overall:

This is an interesting TEM-based study of the 2D MOF, specifically Hf-MOL. The ways/methods that authors used to gain insight into the Hf-MOL formation are good. But the conclusions they are making with the results/observations in-hand are not properly supported. More results and clarifications are needed before this manuscript would be acceptable for publications.

Fundamental questions that need justification:

1. It is not clear that the in-situ TEM observations that authors presented in this work is the natural formation of the Hf-MOL without non-negligible role from e-beam of TEM. More results are

needed to eliminate possible influential role of e-beam of TEM.

[Re] Thanks for raising this important question. The electron beam effect is a common issue that needs to be understood and controlled for many liquid electron microscopy experiments. Here, we can confirm our in-situ observation of Hf-cluster formation and assembly without electron-beam effect by the following reasons. Firstly, we use a small dose rate electron beam to track the cluster formation and assembly ($39.5 \text{ e}^- \cdot \text{\AA}^{-2} \cdot \text{s}^{-1}$ for cluster formation and $130 \text{ e}^- \cdot \text{\AA}^{-2} \cdot \text{s}^{-1}$ for cluster assembly). During the whole process, we don't observe the morphology change of these clusters as can be seen from movie S2, which infers that the structure of clusters wasn't damaged. Secondly, the irradiation species, including the solvated electron (e^-), hydrogen radicals ($\text{H}\cdot$), hydroxyl radicals ($\text{HO}\cdot$), O_2 and H_2 , formed under electron irradiation cannot cause the clusters' formation and assembly^{5,6}. To better understand the electron beam effect in our system, a control experiment of the cluster growth solution exposed to high intensity of electron beam was performed. When we use a dose rate of $920 \text{ e}^- \cdot \text{\AA}^{-2} \cdot \text{s}^{-1}$, what we observed is the formation of HfO_2 nanoparticles rather than Hf-cluster in solution, as shown in Figure R4. Therefore, we conclude that irradiation species have an impact on the structure change/damage of the clusters or the formation of Hf-related nanocrystals, but they would not cause the clusters' formation and assembly.

Figure R4. HRTEM image and corresponding FFT image of the red square region. The distances of the spots in the FFT images match with the (111), (020) and (220) of HfO_2 . HfO_2 was formed within liquid cell at a dose rate of $920 \text{ e}^- \cdot \text{\AA}^{-2} \cdot \text{s}^{-1}$. Scale bar 5 nm.

Specific questions that need clarification:

1. For the results based on Sup. Figure 4 and Movie 1: When liquid is injected into the chip, due

to huge mass difference between the chip and liquid, I would suspect that liquid will rapidly cool down. Do authors know the temperature of the liquid when it was observed in the TEM? Also, how long did it take from “injection” to “imaging” it in TEM? Since it takes some time, due to injection, vacuum check and finding window/focusing in the TEM (5-10 min perhaps), wouldn’t authors expect that formation of the Hf clusters occur before that 3.5 min video was taken. Please provide clarifications about Hf cluster formation kinetics in this context.

[Re] Thanks for the comment. It takes around an hour from "injection" to "imaging" the liquid sample under TEM including the liquid injection and vacuum check. When the liquid cell was loaded into the TEM chamber, the temperature of the tiny liquid within the liquid cell is close to the temperature of the TEM room. We don't rule out the possibility that some clusters formed before the imaging process. From the initial image at 0.0 s (Supplementary Fig. 5), few Hf-clusters were found. This means that Hf-cluster formation occur before the video was taken. In fact, our in-situ results only show that the cluster density increases in the same viewing region.

2. The movie S2 looks like TEM sample drift. Can authors show that it indeed shows clusters moving and not TEM sample drift. If it is indeed Hf cluster moving, why all clusters are moving in same direction synchronous? Wouldn’t they move randomly? Clarification is needed.

[Re] The movement of clusters in the same direction synchronous is caused by the TEM sample drift. We did alignment for movie S2, and the updated video is attached in the revised supporting information. Some frames were selected to show the slow movement of Hf-clusters (Supplementary Fig. 7), because of the strong attachment to the viewing window. We focus the electron beam to generate bubbles in the vicinity of regions with clusters, thereby pushing liquid to flow to increase the movement of clusters, as shown in movie S3.

Supplementary Fig. 7 Sequential TEM images showing the slow motion of clusters. Three clusters are selected to show clusters get close after 720 s. Scale bar 20 nm.

3. In the discussion to Fig. 2b-d, authors show that crystal core formed at the “center” and disorder is at the edges/surface. The “center” of view is the region where there is more effect from e-beam of TEM. Can authors prove that, what they see, is not an effect of TEM e-beam, but natural Hf-MOL formation.

[Re] We can prove the structure in Fig. 2b-d is not an electron beam effect through the following reasoning:

1) From the stability test of Hf-MOL under electron beam irradiation, we know that the Hf-MOL structure can be well maintained at an electron dose rate of $1750 \text{ e}^- \cdot \text{\AA}^{-2} \cdot \text{s}^{-1}$ for at least 49.5 s at room temperature (movie S4). The electron dose rate used for the MOF characterization in Fig. 2b-d is $1700 \text{ e}^- \cdot \text{\AA}^{-2} \cdot \text{s}^{-1}$ at liquid N_2 temperature, and the whole imaging process only lasts for 2 s. Therefore, our dose rate is lower and exposure time

shorter than the dose applied for the stability test, and we expect further preservation of the structure when imaging at cryogenic temperatures.

- 2) A long-time electron beam exposure experiment was performed to understand the damage process of Hf-MOL at an electron dose rate of $1750 \text{ e}^- \cdot \text{\AA}^{-2} \cdot \text{s}^{-1}$ (Supplementary Fig. 14). We found that the Hf-clusters get connected and form an amorphous region under the beam irradiation. This is different from the disorder surface structure in Fig. 2b-d. In Fig. 2d, we confirm that the clusters keep a certain distance of 2 nm rather than get connected with neighboring clusters.

Supplementary Fig. 14 Stability test of Hf-MOLs under electron-beam irradiation at room temperature in TEM mode. a Sequential TEM images and **b** corresponding FFT images showing MOL under electron beam irradiation at dose rate of $1750 \text{ e}^- \cdot \text{\AA}^{-2} \cdot \text{s}^{-1}$ for over 10 mins. Scale bars: a, 10 nm; b, 1 1/nm.

I would guess that in the liquid, there is no “center”, and all Hf-MOL formations should happen randomly in all places.

[Re] The solution can normally consider as a homogeneous system, and Hf-MOLs formations should happen randomly in all places. However, the concentration of clusters in the local area is not the same. There are some reports showing the local concentration difference of building blocks in solution during the nucleation and growth process⁷. Positions with a high concentration of building blocks will preferentially become nucleation and growth site^{8,9}. In our system, the liquid flow can increase the local density of clusters, which can serve as the nucleation site, namely, the "center".

4. There results authors presented, doesn't support the claim (on line 153): “Our results show that clusters first form in solution and are then complexed with ligands to form amorphous solids, followed by the arrangement of the cluster-ligand complex into a crystalline film (Fig. 2e).” The results don't show/say anything about ligands. It is not even clear if there are ligands there. At this point this is only speculation, which might not be true.

[Re] The claim on line 153 in the previous manuscript is made based on the in-situ and ex-situ TEM characterization results. Although we cannot directly image the ligands, we can obtain information about the ligands from the distances between clusters. As can be seen from the high-resolution images and the corresponding inverse fast Fourier transformation (IFFT) images, the distance between adjacent clusters is 2 nm, which is in good agreement with the length of the ligands. Disordered regions containing cluster dimers, trimers and multimers with the same 2 nm distance between adjacent clusters (Fig. 2d) indicate that the clusters complexed with ligands to form non-crystalline solids.

5. Line defects in 3D or 2D crystals has distinct definition. The observation in Fig 3a-b, doesn't constitute line defect and needs to be explained better.

[Re] Thanks for pointing this out. We understand that line defect has distinct definition in the crystalline structure. A line of missing clusters was used in the revised manuscript to replace the description of line defects.

6. You cannot have “amorphous striped morphology” by definition. If there are “strips”, it is not amorphous. Better description is needed.

[Re] We have deleted the structure description of "amorphous striped morphology" for Fig. 3e. We agree with the reviewers' comments that the striped morphology is not amorphous. The striped

morphology was caused by the interlock stack packing of different layers. Therefore, we use interpenetrated layers in the revised manuscript rather than amorphous striped morphology.

Technical

1. In the abstract, it reads "...clusters, point and line defects, dislocations, loop and flat surface terminations, and ...". Dislocation is a line defect. Remove "dislocation" or "line defect".

[Re] We totally agree with this comment. The point and line defects have been replaced by missing clusters in the revised manuscript.

2. There is a recent paper in Chem Mater (S. Ghosh et al (2021)) that has comprehensive discussion on e-beam damage of MOF in TEM.

[Re] We thank the referee for providing this valuable reference which states two distinct stages of structural and chemical changes were identified in ZIF-based MOF under electron beam irradiation. We have added this reference in the manuscript accordingly.

3. In Supplementary figure 1, could authors re-check their files? Low-mag "120 min" image in panel (b) appears to have different magnification (based on size of wrinkled edges), even the scale bars are the same size. Also TEM operational conditions and e-beam doses should be indicated here (for panel b and c) to be sure there is no beam-damage issues here.

[Re] We have rechecked Supplementary Fig. 1 and confirmed that the scale bars are correct in the previous manuscript. In fact, the scale bar length of the image "120 min" is different compared with the other two images in panel b. We have edited the size of the image in panel b in the new manuscript. The imaging conditions are shown in the caption of supplementary Fig. 1 in the revised supporting information file.

4. Figure 1(a) need labeling of the atoms. In the models, which are Hf, C and O atoms? The models also need to be presented from other crystallographic orientations to have full picture of the structure.

[Re] We thank the referee for pointing this out. We mentioned in the caption of Figure 1 that the Hf, O and C atoms are represented by red, blue and black spheres. To make it easier to understand the structural model, we have also added the atoms labelling in the model of Figure 1a in the revised manuscript.

Here, 2D Hf-MOLs were used for high-resolution characterization, so we only list one crystallographic orientation. We agree with the referee's comment to present other crystallographic orientations besides [001] viewing direction to have a full picture of the structure. Following this

suggestion, 3D Hf-MOFs were synthesized by adjusting the ratio of H₂O and HCOOH^{10,11}. The low- and high-magnification TEM images of the 3D-MOFs are shown below. More crystallographic orientation models of these 3D Hf-MOFs are displayed (Supplementary Fig. 2).
12,13

Supplementary Fig. 2 Structure characterization and atomic models of 3D Hf-MOFs. **a** Low- and **b** high-magnification TEM images of 3D Hf-MOFs. **c** Atomic models of 3D-MOFs viewing along [010], [001] and [011] directions. Scale bars: **a** 1 μm , **b** 10 nm.

5. Proper (earlier) reference to Z-contrast ADF-STEM are: S.J. Pennycook, L.A. Boatner, Nature 336 (1988) 565 and S.J. Pennycook, Ultramicroscopy 30 (1989) 58., and not the reference [41] cited in this paper.

[Re] We thank the referee for pointing this out. We have updated these references in the revised manuscript.

6. The sentence (on line 123) “Typically, Hf-MOL formation takes place under solvothermal conditions at temperature.” is not complete.

[Re] We have completed this sentence in the revised manuscript.

7. In the caption to Fig. 2, “movie S1” should be replaced with “movie S3”.

[Re] Movie S1 has been replaced with movie S3 in the caption to Fig. 2.

8. In Fig. 3, labels a-d are missing.

[Re] We have added the labels a-d in the new Fig. 3.

Reviewer #3 (Remarks to the Author):

The work described in the manuscript entitled "Observation of Formation and Local Structures of Metal-Organic Layers via Advanced Electron Microscopy" is significant for the rapidly developing cross-field of electron microscopy for the beam-sensitive materials investigations. Together with this, it gives a unique insight on the formation mechanisms of metal-organic materials. To the knowledge of the reviewer, there are no similar works pushed at the moment. I would recommend this manuscript for publication after the minor revision.

1. In the past 5-7 years the work on the TEM for the metal-organic materials was very intense and I would recommend adding the references to more papers published by now - for example, on TEM, on STEM, on iDPC. It could give a better understanding for the reader about the current field state and the place of this work.

[Re] We thank the referee for his/her highly positive comment on our work, and his/her efforts to make the introduction part more insightful and to make the manuscript overall more thoughtful and readable. After taking the suggestion, we have added the references to more papers related to metal-organic materials characterization using TEM, STEM and iDPC into the introduction part (highlighted on page 3).

2. In general in the text the superior words, such as unique, for example, are a bit overused. I would recommend aligning the manuscript with the more accurate wording.

[Re] We thank the referee's comment and suggestion. We have rechecked and aligned the manuscript with a more accurate description to replace the superior words. Words (e.g., advanced, unique, first time) have been deleted in the revised manuscript.

3. Adding simulated images and overlapping the structure models to the experimentally recorded images would significantly improve the quality of work and the discussion.

[Re] We fully agree with the referee's suggestion of adding simulated images and overlapping structures. We have added more structural models to match the TEM/STEM results (e.g., Supplementary. Fig 9, 10).

References:

- 1 Chen, P. *et al.* Machine-learning-guided morphology engineering of nanoscale metal-organic frameworks. *Matter* **2**, 1651-1666 (2020).
- 2 Li, Y. *et al.* Cryo-EM Structures of Atomic Surfaces and Host-Guest Chemistry in Metal-Organic Frameworks. *Matter* **1**, 428-438, doi:10.1016/j.matt.2019.06.001 (2019).
- 3 Peet, M. J., Henderson, R. & Russo, C. J. The energy dependence of contrast and damage in electron cryomicroscopy of biological molecules. *Ultramicroscopy* **203**, 125-131 (2019).
- 4 Ghosh, S. *et al.* Two distinct stages of structural modification of ZIF-L MOF under electron-beam irradiation. *Chem. Mater.* **33**, 5681-5689 (2021).
- 5 Hill, M. & Smith, F. Calculation of initial and primary yields in the radiolysis of water. *Radiat. Phys. Chem.* **43**, 265-280 (1994).
- 6 Schneider, N. M. *et al.* Electron–Water Interactions and Implications for Liquid Cell Electron Microscopy. *The Journal of Physical Chemistry C* **118**, 22373-22382, doi:10.1021/jp507400n (2014).
- 7 Loh, N. D. *et al.* Multistep nucleation of nanocrystals in aqueous solution. *Nat. Chem.* **9**, 77-82, doi:10.1038/nchem.2618 (2017).
- 8 Zhang, K.-Q. & Liu, X. Y. In situ observation of colloidal monolayer nucleation driven by an alternating electric field. *Nature* **429**, 739-743 (2004).
- 9 Zhang, T. H. & Liu, X. Y. How does a transient amorphous precursor template crystallization. *J. Am. Chem. Soc.* **129**, 13520-13526 (2007).
- 10 Chen, P. *et al.* Probing surface structure on two-dimensional metal-organic layers to understand suppressed interlayer packing. *Nano Res.* **13**, 3151-3156, doi:10.1007/s12274-020-2986-3 (2020).
- 11 Cao, L. *et al.* Self-Supporting Metal–Organic Layers as Single-Site Solid Catalysts. *Angewandte Chemie International Edition* **55**, 4962-4966 (2016).
- 12 Cavka, J. H. *et al.* A new zirconium inorganic building brick forming metal organic frameworks with exceptional stability. *J. Am. Chem. Soc.* **130**, 13850-13851, doi:10.1021/ja8057953 (2008).
- 13 Jakobsen, S. *et al.* Structural determination of a highly stable metal-organic framework with possible application to interim radioactive waste scavenging: Hf-UiO-66. *Phys. Rev. B* **86**, 125429 (2012).

REVIEWERS' COMMENTS

Reviewer #1 (Remarks to the Author):

In the revised manuscript, X. Peng, et al., have carefully addressed the comments raised by the reviewers. The systematic characterizations of the MOF using multiple TEM techniques should be valuable for related communities. Therefore, I recommend it for publication. There are two minor comments for revision:

1. I suggest putting Fig. R1 to supplementary. It is a good information to clarify that the reconstruction from a 800 e/A² dose at 80 kV does not cause obvious crystalline degradation. However, I would argue that higher dose-rate causes less damage than a lower dose rate. The diffraction signal in the power spectrum from 800 e/A² is higher than that from the sum from four reconstructions with 200 e/A² exposure, but it may be caused by some uncertainties from ptychographic reconstructions. For example, during the probe deconvolution procedure, there may be a defocus change from different areas even close-by. The reconstruction from noisier diffraction data from a lower dose may have higher uncertainties.
2. The references for abTEM and Prismatic should be different. It seems that ref [60] is only for Prismatic. Please check.

Reviewer #2 (Remarks to the Author):

In the revised manuscript and in the responses to reviewers' file the authors reasonably addressed all of my concerns. Now I can be positive about publishing the manuscript in Nature Communications.

Reviewer: 1

In the revised manuscript, X. Peng, et al., have carefully addressed the comments raised by the reviewers. The systematic characterizations of the MOF using multiple TEM techniques should be valuable for related communities. Therefore, I recommend it for publication. There are two minor comments for revision:

1. I suggest putting Fig. R1 to supplementary. It is a good information to clarify that the reconstruction from a 800 e/A² dose at 80 kV does not cause obvious crystalline degradation. However, I would argue that higher dose-rate causes less damage than a lower dose rate. The diffraction signal in the power spectrum from 800 e/A² is higher than that from the sum from four reconstructions with 200 e/A² exposure, but it may be caused by some uncertainties from ptychographic reconstructions. For example, during the probe deconvolution procedure, there may be a defocus change from different areas even close-by. The reconstruction from noisier diffraction data from a lower dose may have higher uncertainties.

[Re] Thanks for the comment and suggestion. During the optimization of the imaging conditions for low-dose ptychography, we found that overall higher resolution signal is present if the data was recorded in a high dose-rate regime. To corroborate this finding, we collected 4D-STEM data with a single exposure of 800 e⁻·Å⁻² with four consecutive exposures of 200 e⁻·Å⁻² of adjacent regions, such that the total dose was the same, but the dose rate was different. We observed that for the high dose-rate, the diffraction signal reached an overall higher resolution.

4D-STEM datasets recorded with Nyquist-sampled scan grid allow to analyze the maximum achievable diffraction signal by analyzing the G-function without performing a full phase retrieval. The G-function is an intermediate 4D function in the single-sideband reconstruction and defined as

$$G(\mathbf{k}, \mathbf{K}) = |A(\mathbf{k})|^2 \delta(\mathbf{k}) + A(\mathbf{k})A^*(\mathbf{k} + \mathbf{K})T(-\mathbf{k})^* - A^*(\mathbf{k})A(\mathbf{k} - \mathbf{K})T(\mathbf{k})$$

It can be calculated from the 4D-STEM data by a 2D Fourier transform along the scan coordinate axes.

Using the double-overlap region

$$\mathcal{K} = \{\mathbf{q} : (|\mathbf{q}| < k_0) \wedge (|\mathbf{q} + \mathbf{q}| > k_0) \wedge (|\mathbf{q} - \mathbf{q}| < k_0)\}$$

we can calculate maximum possible diffraction signal of a certain spatial frequency as

$$\left(\sum_{\mathbf{q}' \in \mathcal{K}} |G(\mathbf{q}, \mathbf{q}')| \right)^2$$

This represents the maximum intensity in a certain spatial frequency q that can be achieved if all pixels on the detector are in phase, i.e., if the aberrations were solved perfectly.

We plot the logarithm of the radial integral of this signal below to demonstrate that diffraction information is preserved better for the Hf-MOL sample in a high-dose regime.

Figure R1. A single high-dose-rate exposure produces higher signal than many low-dose-rate exposures

2. The references for abTEM and Prismatic should be different. It seems that ref [60] is only for Prismatic. Please check.

[Re] Thanks for the comment. We have changed the references.

Reviewer #2

In the revised manuscript and in the responses to reviewers' file the authors reasonably addressed all of my concerns. Now I can be positive about publishing the manuscript in Nature Communications.

[Re] We thank the referee for his/her highly positive comment on our work.